

# The Effect of Alternative Seismotectonic Models on PSHA Results – a Sensitivity Study for the Case of Israel

Avital, Matan[1], Davis, Michael[2], Dor, Ory[2], and Kamai, Ronnie[3]

[1]Department of Geological and Environmental Sciences, Ben-Gurion University of the Negev, Beer-Sheva, 84105, Israel
[2]Ecolog Engineering, Inc. Rehovot, 7670203, Israel
[3]Department of Structural Engineering, Ben-Gurion University of the Negev, Beer-Sheva, 84105, Israel

*Correspondence to*: Ronnie Kamai (rkamai@bgu.ac.il)

## Abstract

We present a full PSHA sensitivity analysis for two sites in southern Israel – one in the near-field of a major fault system
and one farther away. The PSHA analysis is conducted for alternative source representations, using alternative model parameters for the main seismic sources, such as slip-rate and $M_{max}$, among others. The analysis also considers the effect of the Ground-Motion Prediction Equation (GMPE) on the hazard results. In this way, the two types of epistemic uncertainty – modelling uncertainty and parametric uncertainty are treated and addressed. We quantify the uncertainty propagation by testing its influence of the final calculated hazard, such that the controlling knowledge gaps are identified and can be treated in future
studies. We find that current practice in Israel, as represented by the most current version of the building code grossly underestimates the hazard, due to a combination of factors, including source definitions as well as the GMPE used for analysis.

## 1   Introduction

Israel lies on an active plate boundary – with the Dead-Sea Transform (DST) separating the African plate on the west from the Arabian plate on the east. According to the historical, biblical, and archaeological records (Ben-Menahem, 1991), devastating
earthquakes with recurrence intervals of approximately 100 years are responsible for the repeated destruction of cultural centres in this region. While Israel benefits from a relative wealth of historical, geological and paleoseismological dataset that can supports Seismic Hazard Assessments (SHA), its instrumental catalogue is poor due to the combination of its young age, sparse spatial coverage, and moderate seismicity rates. Therefore, the current state-of-practice for conducting seismic hazard analysis in Israel suffers from some significant knowledge gaps and methodological shortcomings, which may lead to erroneous hazard
estimations.

The purpose of this study is to quantify the sensitivity of the calculated hazard to the underlying uncertainty in the source and path representations. By that, we intend to contribute to regional SHAs by highlighting, quantifying and ranking the main sources of uncertainties in the calculations. We conduct the analysis for two sites in southern Israel – site #1 is in close proximity to the DST (~20km) while site #2 is farther away (~70km). Specifically, we will explore the sensitivity to:

a)   Alternative seismotectonic models and alternative representations of the DST faults

b)   Segmentation of the main seismic sources

c)   Uncertainty in input parameters, such as slip-rate, activity-rate, and maximum magnitude

d)   Alternative Ground Motion Prediction Equations (GMPEs)



## 1.1 Components of Uncertainty

In Probabilistic Seismic Hazard Analysis (PSHA), uncertainty can originate from three main sources – the seismic source, the propagation path, and the site response. Uncertainties are propagated throughout the analysis and have been shown to dominate the results for high-risk projects, such as Nuclear Power Plants (NPP), hydraulic dams, and major lifelines (e.g., Abrahamson and
Bommer, 2005; PateCornell, 1996; Rodriguez-Marek et al., 2014). It is common to describe uncertainty as either aleatory or epistemic (e.g. Abrahamson and Bommer, 2005; Paté-Cornell, 1996). *Aleatory* uncertainty describes the inherent variability in a physical process, one which cannot be fully explained by the currently proposed physical model, also simply called randomness. *Epistemic* uncertainty is the scientific uncertainty in the model or the underlying parameters. It can result from lack of knowledge or insufficient collected data, and hence could generally be reduced by some amount of effort or monetary resources.
The epistemic uncertainty can be further divided into *Modelling* uncertainty, representing alternative simplified representations of the actual physical process and *Parametric* uncertainty, representing the uncertainty in the value of the model's input parameters (e.g. Abrahamson et al., 1990; Toro et al., 1997). Modelling uncertainty represents the differences between the actual physical process that is being modelled, and the simplified model which is used to predict the response. In this study, we focus on the epistemic uncertainty (both modelling and parametric), related with the seismic source and propagation path, for a PSHA
analysis of two sites in southern Israel.

## 1.2 Seismic Hazard Practice in Israel

The most recent update to the Israeli building code (SII, 2013) and its associated seismic-hazard map (Klar et al., 2011) is considered herein to represent the state-of-practice of seismic hazard analysis in Israel. This practice will be further related to herein as the 'SI413' model. The underlying seismotectonic model in SI413 is shown in Figure 1. It is composed of areal sources
only, based on the work of Shamir et al. (2001). The activity rates within the seismic zones were defined based on the uniform earthquake catalogue, constructed from combined historical and instrumental data (Shapira and Hofstetter, 2002; Shapira et al., 2007a). The seismic zones are all assigned a truncated exponential (TE) magnitude-frequency distribution (MFD), as typical for areal sources (Cosentino et al., 1977). Finally, the horizontal spectral acceleration predicted by the map are calculated using the Campbell and Bozorgnia (2008) GMPE, originally developed for California and the Western US.
The underlying assumptions used to construct the SI-413 are obsolete. They are approximately 20 years behind current world-practice in PSHA, especially considering the extensive geological and geodetic research performed on the DST faults in the past 30 years. Some of the main limitations in the SI413 model are specified and explained below:

    (a)  All seismic sources within SI413 are represented as areal source zones (ASZs) rather than planar sources. Gulerce and Vakilinezhad (2015) show that hazard estimates, especially for near-fault sites, are significantly and systematically
smaller when using areal source zones to represent major seismic sources, rather than using linear fault models in the PSHA.

    (b)  Large areas are left outside of defined ASZs (as shown in Figure 1), resulting in their seismic activity rates in the PSHA to be defined as zero. This is typically not allowed in hazard studies, because the possibility of an earthquake occurrence can never be completely rejected, even in a previously inactive region. Therefore, some minimal background seismicity
has to be accounted for in places where there are no mapped seismic sources.

    (c)  All earthquakes in SI413 are represented as point-sources. While this may be reasonable for small-to-moderate earthquakes (M≤6), it is clearly wrong for larger earthquakes which occur along rupture planes where rupture length may be at lateral dimensions similar to the affected zone. This representation is especially significant for the distance calculations within the GMPE, because most recent GMPEs use some sort of rupture distance (e.g. $R_{rup}$, $R_{JB}$). For





example – consider two sites that are 200 km apart from each other, but are at two ends of a major fault. These two sites could be at a very short rupture distance from a large earthquake, but at much longer distance (~100 km) if the source is represented as a point source in a middle location. In such a case, the calculated hazard would be much smaller. This difference is further emphasized in Israel where the country's shape – long and narrow - lies parallel to the DST fault system.

(d) The Magnitude-Frequency Distribution (MFD) of all seismic sources in SI413 is the truncated-exponential (TE), also known as the Gutenberg-Richter (1944) model, which is the most commonly used model for ASZs or places in which good characterization of the seismic sources is unavailable. However, this relation is found to underestimate the occurrence rates of large earthquakes in regions dominated by large faults. An alternative model is the characteristic model (Schwartz and Coppersmith, 1984), or, preferably, the composite model (Youngs and Coppersmith, 1985), that combines the two such that 94% of the seismic moment is released by large characteristic earthquakes and only 6% of the moment is released by the exponential 'tail'. Other versions of the composite model also exist. For instance, the Uniform California Earthquake Rupture Forecast, Version 3 (UCERF3, Field et al., 2014) adopted a composite Characteristic Magnitude MFD, by allowing the characteristic part to account for two-thirds of the seismic moment, with the Gutenberg-Richter accounting for only a third.

(e) The activity rates used in SI413 are based on combined historical and instrumental data (Shapira and Hofstetter, 2002; Shapira et al., 2007b). The recorded seismicity data includes barely 20 years in which the catalogue is considered complete for M≥2.0. These rates are equivalent to a slip rate of approximately 1 mm/year, which is significantly lower than geological and geodetic estimates, as shown and discussed later.

(f) Maximum magnitudes are mostly based on historical estimates (e.g. Ben-Menahem, 1991). Instead, it is more common in recent PSHA studies to employ global empirical relationships to estimate the physical constraints on the maximum magnitude based on physical fault dimensions (e.g. Wells and Coppersmith, 1994)

Following these limitations, Davis and Dor (2014) proposed an alternative seismotectonic model, presented in Figure 2. This model was developed by adaptation of principles that are currently in use by national/country PSHA models such as UCERF3 in California (Field et al., 2014), SHARE in Europe (Woessner et al., 2015), and J-SHIS in Japan (Fujiwara et al., 2006); it will be further referred to as the 'DD14' model. The DD14 model represents the main DST faults, as well as the Carmel fault, as linear source zones. The model also includes Fault-Zone Polygons (FZP) surrounding the linear seismic sources, and background seismicity polygons from the Shamir et al. (2001) model, representing off-fault seismicity. In this model, large earthquakes ($6.5 \leq M \leq M_{max}$) occur on the linear sources, while small to moderate earthquakes ($M_{min} \leq M < 6.5$) are represented as point-sources within the FZP. The seismic moment on the main seismic sources is balanced between the two components of fault representation as follows: a truncated-exponential MFD is used to represent the FZP with the calculated activity rates based on the seismic catalogue (Shapira and Hofstetter, 2002) while a characteristic-earthquake MFD is used to represent the linear sources, using the geological estimates of slip rate, after subtracting the seismic moment released by the FZP. The off-fault polygons are identical to their equivalent in the SI413 model.

## 1.3    Previously Published Hazard Studies in Israel

A comprehensive source characterization study was performed by the Israel Electric Corporation Ltd. (IEC, 1993, 2002) for the Shivta-Rogem site in the western Negev desert (site 2 in our analysis), which was identified as a potential site for a nuclear power-plant in the mid -1980's. As part of the source-characterization study, extensive field-work was performed, and four





additional capable faults were identified in the site region – the S-19, Zin, Sa'ad-Nafha, and Ramon faults. These faults were assigned activity rates, including acknowledgement of the associated uncertainty. These additional faults are not included in the analysis presented in this paper.

A hazard sensitivity study for the Shivta-Rogem site (site 2 in our analysis), was conducted by Rabinowitz et al. (1994), using the multi-parameter approach (Rabinowitz and Steinberg, 1991). In their analysis, Rabinowitz et al. considered only two seismic sources, both near the site; The DST fault system was not considered. Their main outcome was that the hazard calculations were more sensitive to the activity rate of the Zin fault than to its exact dimensions and associated maximum magnitude.

Two recent papers (Al-Tarazi and Sandvol, 2007; Haas et al., 2016) use the gridded-seismicity approach (Frankel, 1995) to produce hazard maps for the entire DST region, based on recorded and historical seismic catalogues, without defining any linear or areal source zones. This approach is becoming more common in areas in which the seismic sources are undefined, or for representation of background seismicity, but is inappropriate for representing large known mapped faults, such as the DST (e.g. Pecker et al., 2017). We do not consider gridded seismicity in this study, although we believe it should be the approach for future definition of off-fault seismicity in our region.

## 2 Defining the Range of Epistemic Uncertainty

In order to systematically explore the hazard sensitivity to the uncertainty associated with different input parameters, we define six models, gradually adding or changing components, as outlined in Table 1, and detailed below:

**Table 1. List of all seismotectonic models used for analysis in this study and their main features**

| Model No. | Source geometry for known mapped faults | MFD for main seismic sources | Including near-site sources | Includes Parametric epistemic uncertainty | Includes Segmentation | Comments |
|---|---|---|---|---|---|---|
| 1 | Areal sources ('polygons') only | TE | - | - | - | SI413 (Figure 1) |
| 2 | Linear fault + FZP | TN on faults + TE in FZP | - | - | - | DD14 (Figure 2) |
| 3 | Linear fault | YC | - | - | - | |
| 4 | Linear fault | YC | Yes | - | - | |
| 5 | Linear fault | YC | Yes | Yes | - | |
| 6 | Linear fault | YC | Yes | Yes | Yes | |

FZP – Fault Zone Polygon, TE – Truncated Exponential, TN – Truncated Normal, YC – Youngs and Coppersmith composite model

*Model 1*

This model is based on the SI413 model (presented in Figure 1), as explained above.

*Model 2*

This model is based on DD14 (Davis and Dor, 2014), as explained above.

*Model 3*

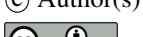



In this model, the same six mapped faults as in Model 2 are represented as linear sources only, without a FZP. All earthquakes – small and large – occur on the fault trace. The MFD is the composite model (Youngs and Coppersmith, 1985) - called herein YC for brevity - which allows the seismic moment to be distributed between the characteristic part (accounting for 94% of the moment release) and the TE part (accounting for 6% of the moment release). All other seismic sources are left identical to their

representation and characterization in SI413.

Note that there is an inherent inconsistency between source representation in Models 2 and 3: in Model 3 the FZP seismicity-based activity rates are eliminated, in favour of MFDs fully represented by geological estimates of slip-rates. Another inconsistency stems from the different moment distribution. Figure 3 shows three MFDs based on the same long-term slip-rate, representing Model 1 (TE), Model 2 (line+FZP), and Model 3 (YC). The activity rates of small-magnitude events will always, by

definition, be smaller for the YC MFD, because most of the moment is released in the larger events. This has been extensively discussed by others (e.g. Gülerce and Vakilinezhad, 2015) and will not be repeated here. Defining consistent parameters for the different fault representations is beyond the scope of this paper, because our main focus is the potential hazard sensitivity to these uncertainties and inconsistencies.

*Model 4*

Model 4 is identical to Model 3, including additional near-site seismic sources which are not part of the regional seismotectonic model (see Figure 4). For site #2 these near-site sources include a 20-km radius background-seismicity polygon, as well as an active segment of the Zin fault. For site #1 the near-site source includes only a 12-km radius background-seismicity polygon. The background polygon here is smaller, because it is already very close to the DST segments and a larger radius polygon may lead to double-counting of seismicity which is already associated with the DST. All other seismic sources are left identical to

their representation and characterization in previous models.

*Model 5*

Model 5 deals with the parametric uncertainty in slip-rate of DST segments, activity rates of background polygons, and seismogenic depth, affecting Mmax. It is based on Model 4, but includes the full range of values for each of these input parameters, defined by a comprehensive literature review, as explained in the following section. All other seismic sources are left

identical to their representation and characterization in previous models.

*Model 6*

Model 6 accounts for different segmentation models of the DST, as detailed in the following section. All other seismic sources are left identical to their representation and characterization in previous models.

## 2.1  Parametric Epistemic Uncertainty

### 2.1.1  Slip Rate

The slip rate of the DST fault system has been extensively studied. The slip rate of a given fault can be evaluated using various disciplines, from classical field geology (e.g. trenching), paleoseismology, recorded seismology, geodesy and more. These different research approaches do not only represent different tools, but also different time scales for estimating the rate of motion on the fault - from millions of years in geological studies, to a few years in geodetic studies. It is quite possible that the rate of

relative movement along a complex fault system such as the DST has changed throughout the geological history since the beginning of its activity in the Miocene period, and therefore uncertainty can be significant. Figure 5 summarizes the various assessments of previous studies, separated by discipline.



An analysis of the various estimates of the slip rate along the DST fault system, as shown in Figure 5, shows that the overall range is between 1-20 mm/year, but that estimates higher than 8 mm/year are based on geological studies that represent time-windows of millions of years. Because most of the estimates range from 1-8 mm/year, we decide to take this range as the representative range of epistemic uncertainty for the slip-rate along the DST fault segments.

### 2.1.2 Segmentation

The segmentation model of the DST fault system contains significant epistemic uncertainty, due to the wide range of estimates in the scientific literature (e.g. Garfunkel, 1981; Garfunkel et al., 1981; Gomez et al., 2007). In this study, we focus on two end-members for the segmentation representation: (1) the continuous model, representing both Arava and Jericho faults as single-stranded seismic sources, as shown in Figure 2, and (2) the segmented model, shown in Figure 6, partitioning the Arava and Jericho faults into three segments each. This segmentation is mainly based on the map of active faults, published by the Geological Survey of Israel (Sagy et al., 2013) as well as on the work of Sadeh et al. (2012). The continuous model does not ignore geometrical segmentation of the DST, but rather assumes the likelihood of multi-segment ruptures. Modern seismic hazard models (e.g. UCERF3) relax segmentation assumptions and include multi-segment ruptures as the observation of such fault behaviour becomes more frequent (e.g. Mw=7.3 Landers, 1992; Bray, 2001). In fact, about 40% of mapped ruptures propagated through fault-steps of up to 3-4 km (Wesnousky, 2008).

Table 2 lists the different fault segments in our analysis and their respective lengths. In the Dead-Sea basin itself, Sadeh et al. (2012) suggest two faults on both sides of the basin – eastern and western. In order to maintain the correct moment balance in the segmented model, only the eastern segment was chosen to represent the faulting in the Dead-Sea basin.

**Table 2. Summary of the seismogenic depth estimates for the different DST segments (both for the segmented and continuous models) and their respective $M_{max}$ estimations, using the Hanks and Bakun (2002) empirical relationship.**

| Segment No. | Fault Name | Depth [Km] min | Depth [Km] max | Length [Km] | A [Km^2] min | A [Km^2] max | Mmax ± sigma min | Mmax ± sigma average | Mmax ± sigma max |
|---|---|---|---|---|---|---|---|---|---|
| | Jericho continuous | 11 | 27 | 201 | 2286 | 5377 | 7.45 | 7.82 | 8.18 |
| J1 | Kinnarot Vally | 10 | 23 | 57 | 567 | 1304 | 6.64 | 7.00 | 7.36 |
| J2 | Jericho Vally | 12 | 26 | 64 | 754 | 1661 | 6.81 | 7.15 | 7.50 |
| J3 | Dead Sea East | 12 | 30 | 80 | 963 | 2407 | 6.95 | 7.33 | 7.72 |
| | Arava continuous | 12 | 27 | 191 | 2292 | 5239 | 7.45 | 7.81 | 8.17 |
| A1 | North Arava | 12 | 29 | 90 | 1081 | 2611 | 7.01 | 7.39 | 7.77 |
| A2 | Central Arava | 12 | 27 | 52 | 626 | 1408 | 6.70 | 7.05 | 7.41 |
| A3 | Avrona Fault | 12 | 25 | 49 | 586 | 1220 | 6.66 | 6.99 | 7.33 |

### 2.1.3 Seismogenic Depth

The seismogenic crustal depth is used to define the maximum fault-plane width, assuming that earthquakes do not occur below the seismogenic depth. The depth of the fault is an important parameter because it is used to calculate the maximum/characteristic magnitude ($M_{max}$), using empirical equations that link the rupture area with the expected moment magnitude (e.g. Wells and Coppersmith, 1994). In this study we use the updated version proposed by Hanks and Bakun (2002).





There is a range of estimates for the seismogenic crustal depth along the DST in Israel. In this study we focus on three studies, as shown in Figure 7. Sadeh et al. (2012) used GPS velocities between 1996 and 2008 to infer slip rate and locking depth along the various segments of the DST. Shalev et al. (2013) analysed temperature data from oil and water wells across Israel. They present a cross-section of calculated temperature gradients along the DST. At temperatures below 300-350 °C, the deformation is

expected to be brittle, and hence that range can approximately represent the seismogenic zone. Wetzler and Kurzon (2016) used a local velocity model to relocate the ~15,000 seismic events recorded by the Geophysical Institute of Israel (GII) between 1985 and 2015. Their relocated depths are then analysed to re-estimate the seismogenic depth along the DST. As can be seen in Figure 7, despite significant differences between the three depth profiles, they all agree that there are variations from north to south and that the Dead-Sea basin itself is anomalously deep.

Table 2 lists the depth range obtained for each segment in the DST system, together with the respective $M_{max}$, calculated using Hanks and Bakun (2002). The calculation approach was slightly different for the segmented model and the continuous model, as follows: in the segmented model, each segment was assigned a maximum and minimum depth, according to its location along the profile presented in Figure 7. Based on these end-values, three estimates for $M_{max}$ were obtained – the average value uses an average calculated depth and the median empirical estimate of $M_{max}$. The maximum and minimum $M_{max}$ estimates are calculated

from the depth end-values, as well as adding and subtracting one standard-deviation from the empirical relationships, respectively. In the continuous model, the average depth of the Arava and Jericho faults are calculated using a weighted average of the seismogenic depth, because they are each ~200 km long and the estimated depth varied along their length. Then, the average $M_{max}$ is calculated using the median estimate and the maximum and minimum $M_{max}$ estimates were obtained by adding and subtracting one standard deviation, respectively.

### 2.1.4  Additional Near-Site Sources

The activity rates for the two near-site background polygons was calculated based on the GII catalogue, counting events with M ≥ 2, and considering catalogue completeness (Shapira et al., 2007b). The epistemic uncertainty on the calculated activity rates was introduced by using the 5% and 95% margins of the Weichert (1980) model, which accounts for the possibility that the number of recorded events does not fully represent the true long-term activity of the region. The final activity rates are presented

in Table 3.

**Table 3. Activity rates for the background near-site polygons**

|  | Site #1 | | | Site #2 | | |
|---|---|---|---|---|---|---|
|  | **5%** | **observed** | **95%** | **5%** | **observed** | **95%** |
| **Activity rate** | 0.00036 | 0.0016 | 0.005 | 0.00034 | 0.0016 | 0.005 |

The assessment of the maximum magnitude of an areal source zone, especially one with little recorded seismicity, is quite uncertain.  Two statistical approaches are escribed in Abrahamson et al. (2004) – the 'Kijko' approach (Kijko and Sellevoll, 1989) and the 'EPRI' approach (Johnston et al., 1994). However, because both approaches are based on the recorded seismicity

and because our background polygons only include 4 recorded events with M ≥ 2 each, these approaches are found inappropriate. Therefore, we arbitrarily choose $M_{max}$=6.0 for the median value with ±0.5 magnitude unit to account for the epistemic uncertainty on $M_{max}$.

The Zin fault segment, which is also added as a known active fault in the vicinity of site #2, has been studied by Avni and Zilberman (2006). While there is some uncertainty as to its spatial extension, in this paper we include the mapped active segment

only, which is 2 km long, in our analysis. The Zin fault is assigned a slip rate of 0.003-0.03 mm/year by previous hazard studies




in the region (IEC 2002), which is adopted in this study as well. The maximum magnitude is calculated from the fault dimensions, with a median value of $M_w = 4.7$.

### 2.1.5 GMPE

Despite several attempts to develop a local GMPE for Israel (e.g. Gitterman et al., 1994; Meirova et al., 2008), such attempts led
to models which were poorly constrained at large magnitudes and hence inappropriate for engineering practice. Due to the lack of a local GMPE, the current practice (namely SI413) is to use the Campbell and Bozorgnia (2008) GMPE, called here CB08 for brevity, for hazard calculations. While the CB08 represents the state of the art for the time of its publication, there have been major advancements in the field – both globally and regionally. For example – the Next Generation Attenuation (NGA) project itself, has published a significant update, based on a much wider global dataset and including smaller magnitudes so that scaling
of small-to-moderate events is greatly improved. In addition, even in California, for which these GMPEs have originally been developed, it is common to use more than one GMPE in the analysis, so that modelling epistemic uncertainty is accounted for.
In this study, we test the sensitivity to this parameter, by conducting the analysis with six different GMPEs, as summarized in Table 4. The GMPE uncertainty is included only in Model 5.

**Table 4. The GMPEs used for analysis and their associated Ground-Motion database**

| GMPE | Abbreviation | Ground Motion Database |
|---|---|---|
| Campbell and Bozorgnia (2008) | CB08 | NGA |
| Abrahamson et al. (2014) | ASK14 | NGA‑West2 |
| Boore et al. (2014) | BSSA14 | NGA‑West2 |
| Campbell and Bozorgnia (2014) | CB14 | NGA‑West2 |
| Akkar et al. (2014) | ASB14 | RESOURCE |
| Bindi et al. (2014) | Bindi14 | RESOURCE |

Finally, the logic tree shown in Figure 8 represents the parametric epistemic uncertainty in models 5 and 6 in our analysis.

**3    Hazard Results**

We conduct the PSHA analysis using the Haz45i open-source program (PG&E, 2010. Also on github https://github.com/abrahamson/HAZ). We present the results for two spectral periods – T=0.01sec (referred to herein as PGA) and T=1sec, representing high and low-frequency contributions, respectively. We generally focus on two exceedance rates: (a) 10% in 50 years, corresponding to a return period of 475 years, which is the common hazard level for planning of ordinary
structures, and (b) $10E^{-5}$, corresponding to a return period of 100,000 years, commonly used for highly sensitive facilities, such as nuclear power plants.
The effect of modelling uncertainty is shown in Figure 9, comparing hazard curves obtained from Models 1 through 4, both sites and both spectral periods. The three horizontal lines on the curves represent, from top to bottom, exceedance probabilities of 10%@50yr, 2%@50yr, and $10E^{-5}$, respectively. It is clearly seen, here and in following figures, that the effect of epistemic
uncertainty increases with decreasing exceedance probability. All four panels in Figure 9 show that Model 1 is consistently underestimating the hazard with respect to the other models. At long periods (T=1sec), there is basically no difference between the other models (2 through 4), while differences do exist at short periods (T=0.01sec, PGA). For example – let us begin with site



#2 (Figure 9c), which is in the far-field with respect to the large earthquake generators – the Arava and Jericho segments of the DST. Looking at the low exceedance rate, $10E^{-5}$, which is driven by large magnitudes at long distances (on the DST) – we see a clear increase in the hazard estimate from Model 1 to Models 2 and 3. This is due to changing the source representation from an areal source to a linear source, but mostly due to the associated change in MFD, with the YC distribution giving a much greater

rate for large magnitude events than the TE, as shown in Figure 3. There is a further increase in hazard moving from Model 3 to Model 4, due to the additional background polygon. Despite its very low activity rate (0.0016) and moderate $M_{max}$=6.0, this source adds significant hazard to the site at very low exceedance rates, because all other sources are at much larger distances. In Site #1, however (Figure 9a), there is practically no change between Models 2,3, and 4, because the DST sources are so close, that an additional low-seismicity background polygon does not change the hazard. The only noticeable difference for Site #1 is

observed at short spectral periods (PGA) and relatively high exceedance rates, in which Models 1 and 2 are in fact higher than models 3 and 4. This, again, relates to the difference in MFD, shown in Figure 3: Due to the different moment distribution, small magnitudes get higher rates in the TE models than in the YC MFD, but this typically affects exceedance rates which are well above design levels.

The effect of the segmentation is shown in Figure 10, comparing hazard curves obtained from Models 5 and 6, including all

branches of the logic tree, for both sites and both spectral periods. In all four cases, the segmented model (Model 6) is higher than the continuous model (Model 5) at high exceedance rates, due to the increased probability of a small-to-moderate event occurring on the DST when it is comprised of six instead of three segments. However, the segmented model has a reduced chance of a large earthquake, leading to the segmented model being lower than the continuous model in three out of the four cases (Figures 10a, b, and d). In Figure 10c, corresponding to Site #2 at short spectral periods, the continuous and segmented

models overlap at low exceedance rates. That is because the hazard there is dominated by large earthquakes at short distances, occurring on the background polygon and not on the DST faults (as seen in Figure 9c).

The parametric epistemic uncertainty, shown by the range of hazard curves in Figure 10, is further separated into the different parameters and different seismic sources in Figure 11. In this plot we present Model 5 (continuous) only, at short spectral periods (PGA) only, and at two distinct exceedance rates. The seismic sources are ranked by their contribution to the hazard uncertainty,

such that the most contributing sources are at the top of the plot. The red squares correspond to activity rates of areal sources, the green circles correspond to slip-rates of linear sources, and the blue diamonds correspond to different evaluations of $M_{max}$. It is clearly seen that the hazard in Site #1 is dominated by the nearby DST linear sources (Arava and Jericho), while the local background polygon contributes less to the hazard because it has a smaller activity rate and can generate smaller magnitude earthquakes. Furthermore, within the parametric uncertainty associated with the two DST faults, the slip-rate has a greater effect

on the hazard sensitivity than $M_{max}$, especially for the Jericho fault. The hazard in Site #2 is dominated by the background polygon for both exceedance rates. While the contribution of the DST faults at high exceedance rates is quite substantial, they are practically insignificant at low exceedance rates. Figure 11 also shows that the parameter which contributes most to the hazard uncertainty at Site #2 is the activity rate of the background polygon, which is distinctly more significant than $M_{max}$ of the background polygon, while for the DST segments both $M_{max}$ and slip-rate are almost equally substantial.

The effect of alternative GMPEs is presented in Figure 12, in which all hazard curves are obtained for the weighted average of Model 5, using six different GMPEs (Table 4). The main observation from this plot is that the hazard curve obtained with CB08 has a steeper slope (in the hazard domain) than the rest of the GMPEs. The slope of the hazard curve is related to the aleatory variability, represented both by the number of standard deviations considered in the hazard integral, or by the value of standard deviation (also called Sigma) within the GMPE. In this analysis, all hazard calculations were made using three Std above and

below the median. Therefore, the different slope may be related to the value of Sigma in CB08, which is slightly smaller than the



other models. This is a significant observation, mainly due to the fact that the current Israeli building code, SI-413 is calculated using CB08 alone, which should probably be updated to include a range of more recent models in future developments.

Finally, the overall effect of parametric epistemic uncertainty – in GMPE, $M_{max}$, Slip rate for linear sources and Activity rate for areal sources – is summarized in Figure 13, compiling results from Model 5 for both sites at short spectral periods (PGA) and

two exceedance rates. This plot represents the relative effect of each of the uncertain parameters, with respect to the weighted average of Model 5 using the CB08 GMPE (the solid red line in Figure 10 also shown as a blue solid diamond in this plot), by normalizing each subplot to a reference PGA value, listed within the plot. For each parameter, the median value is shown by a red vertical line, the 25th and 75th percentiles are shown by the box, and the full range of results is represented by the horizontal line. It can be seen that for Site #1, the most significant parameter is the GMPE, followed by slip-rate of the DST faults. For Site

#2 the most significant parameter is the GMPE, followed by activity rate of the background polygon and only then slip-rate of DST faults. While the GMPE effect on hazard ranges between 40% for high exceedance rates in Site #2 to 100% for low exceedance rates in Site #1, the effect of slip-rate or activity rate is only about 20%-40%.

## 4   Discussion and Summary

Some key elements and assumptions in the current practice of SHA in Israel are identified and addressed. A hazard sensitivity

analysis is conducted, while gradually adding components, in order to identify the main controlling uncertainties. The study is performed for two sites – near, and far, from the major seismic source of the region – the DST. The analysis highlights the main shortcomings and limitations of the current national building-code model SI413. Our main conclusions are listed below:

1.  From the parametric uncertainty perspective - the GMPE was found to control hazard uncertainty, followed by slip-rate of the DST for the near-field site and by background activity levels for the far-field site. The maximum magnitude, set by physical

fault dimensions was found to be less significant in terms of hazard uncertainty, although this could possibly be related to the limited range of $M_{max}$ resulting from such physical constraints.

2.  From the modelling uncertainty perspective - we conclude that the combination of assumptions underlying SI413 constructively adds up to underestimate hazard, both near and far from the main regional seismic sources. These modelling assumptions are again pointed out and discussed below:

(a)  The representation of the DST sources as uniformly distributed areal zones, in which all earthquakes occur as point-sources underestimates the distance measures from large ruptures and hence leads to an underestimation of hazard. Large-magnitude earthquakes are preferably represented as long ruptures on linear sources in modern SHA models.

(b)  The seismicity-based activity rates, assigned to the DST faults, are in disagreement with slip-rate estimates from paleoseismic and geological data. This leads to underestimation of seismic moment accumulation on the DST and hence

to additional underestimation of the hazard.

(c)  The Gutenberg-Richter MFD, assigned to the DST sources, has a significantly reduced rate of large magnitude events when compared to other MFDs, such as the composite YC model. While there is no strong evidence for characteristic behaviour of the DST, we believe the available data is insufficient to safely disregard it. For example – Hamiel et al. (2009) analysed paleoseismic, historical, and geodetic data, representing 60,000 years on three different segments of the

DST. They conclude that the Gutenberg-Richter distribution is a stable representation of the seismicity of the DST. However, Hamiel et al. (2009) do not address the inconsistency between observed seismicity and slip rates, as presented in Figure 3, which can be accounted for by applying a composite MFD. Furthermore, the paleoseismic data (which governs their large-magnitude portion of the MFDs) for two of the three DST segments in their analysis uses normal





displacement primarily on rift-margin faults, while large strike-slip events that presumably govern the long-term moment release are, in-fact, not represented in the collected data. In the third segment, paleoseismic data comes from brecciated beds ('seismites') for which the seismic sources cannot be determined. We therefore believe that their distribution better represent the background seismicity along the DST, and that it is statistically insufficient to contradict

the possibility that the DST has characteristic behaviour, similar to what is commonly assumed for large faults in similar tectonic settings (e.g. San-Andreas, North Anatolian Fault). Therefore, we believe that the DST must be represented by a composite model for SHA until safely proven otherwise.

(d)   Hazard in SI413 is calculated using a single GMPE – CB08, which hasn't been sufficiently tested and/or adapted for the region. This GMPE happens to have a relatively low median and standard-deviation, leading to a steeper slope of the

hazard curve with respect to other, more recent, GMPEs. National maps, especially for regions in which a local GMPE does not exist, should always include more than a single GMPE, to better represent epistemic uncertainty in this parameter.

3. The inclusion of background seismicity, even if at very low activity rates, is significant for far-field sites and less significant for near-field sites, in which the proximity to the main faults dominates hazard. At farther sites, such as site #2 herein, the

background seismicity controls the hazard at long return period, suggesting that a layer of background seismicity (preferably using the gridded seismicity approach) should be added into the seismotectonic model of the updated seismic hazard map for Israel. A minimum default level must be set, such that background seismicity cannot be zero at any point, even if the (limited) recorded catalogue hasn't identified an event in that area as yet.

While we are aware that our analysis was conducted for Southern Israel only, the configuration of the DST - set lengthwise the long dimension of the country - suggests that the main conclusions are likely relevant for other regions of the country as well. We suggest conducting similar studies in additional locations in central and northern Israel, but believe that the main difference will be local sources which are not included in the national analysis. We conclude by suggesting, for future updates of the national hazard maps, that epistemic uncertainty will be fully covered and addressed, as done in other developed countries

around the world.

**Acknowledgements**

We thank the Israeli Ministry of Energy and National Infrastructure for funding this research. Norm Abrahamson, Debra Murphy and Christie Hale are greatly thanked for their technical support with the Haz45 program.



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




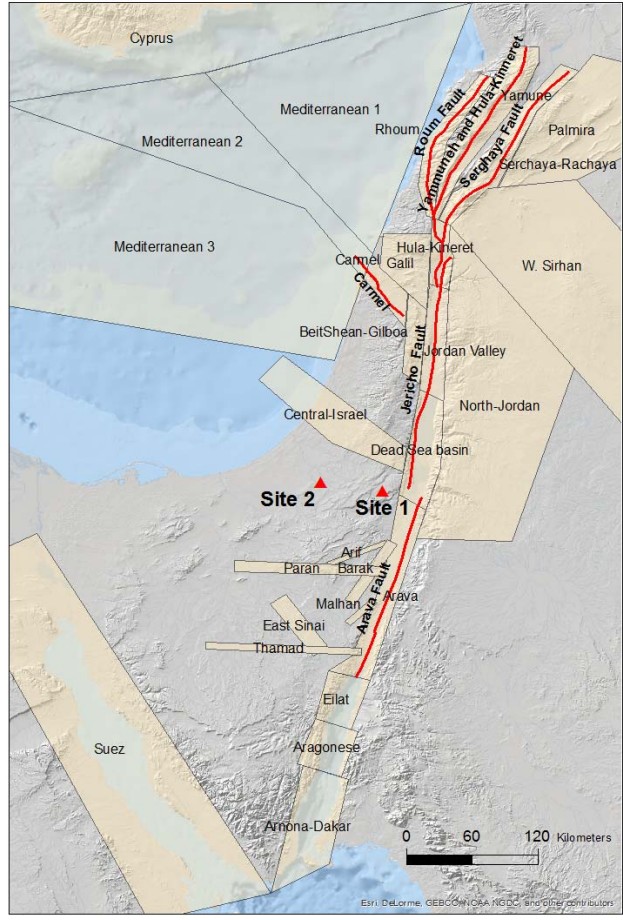

**Figure 2. The seismotectonic model proposed by Davis and Dor (2014), combining linear faults with buffer zones and areal source zones for seismic sources with no defined underlying faults.**




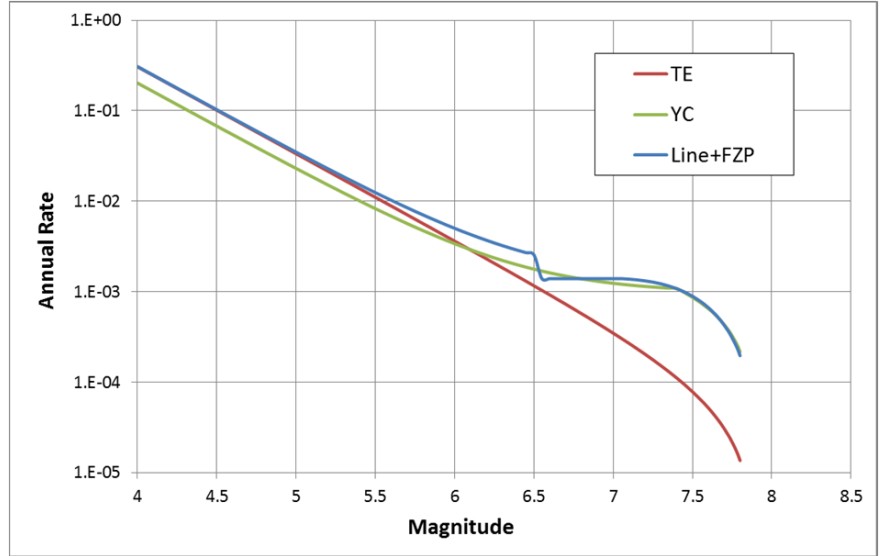

**Figure 3. Example magnitude-frequency distributions for the 191-km continuous Arava Fault segment. TE – equivalent to Model 1, with the associated activity rate of $N(M_{min}=4.0) = 0.3$, which is equivalent to a slip-rate of 0.6 mm/year. Line+FZP – equivalent to Model 2, in which the FZP has the same parameters as in Model 1 and the line has the same parameters as in Model 3. YC – equivalent to Model 3, in which the fault is assigned a composite model with a slip-rate of 3.5 mm/year.**



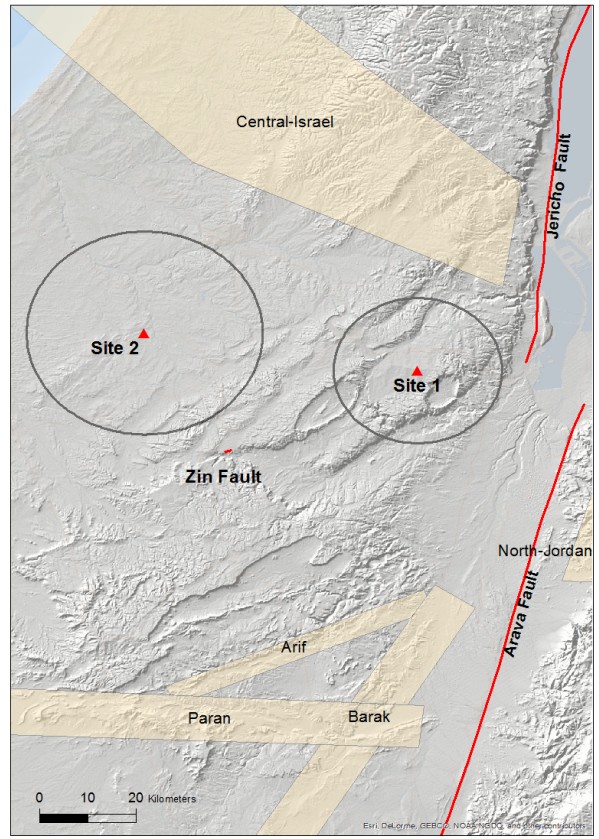

**Figure 4. Model 4, showing the two sites and their respective near-site sources.**





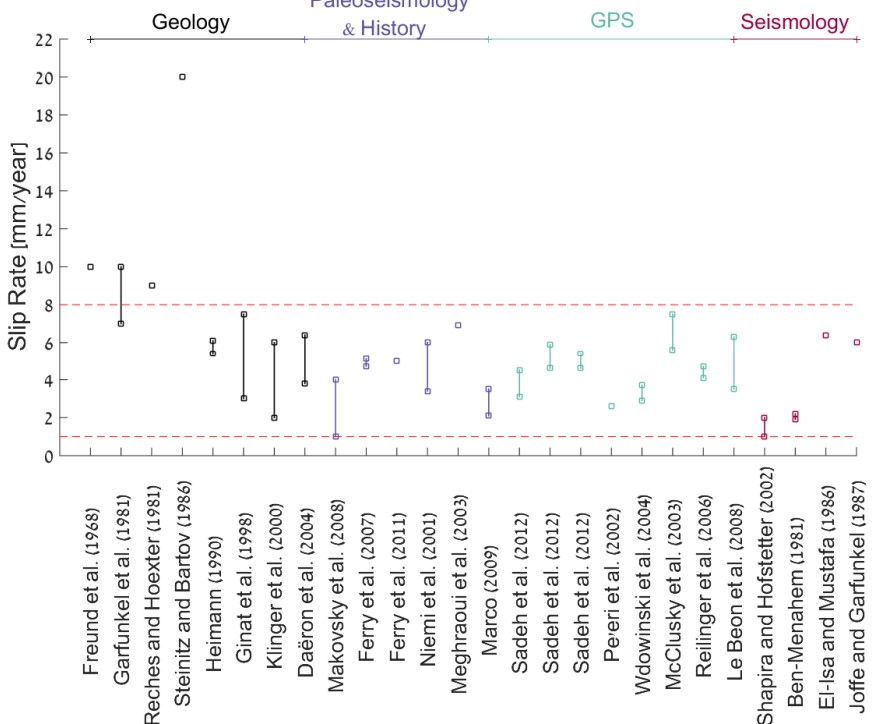

**Figure 5. Summary of the different slip-rate estimates, based on previous studies from a range of disciplines, and representing different time-windows.**





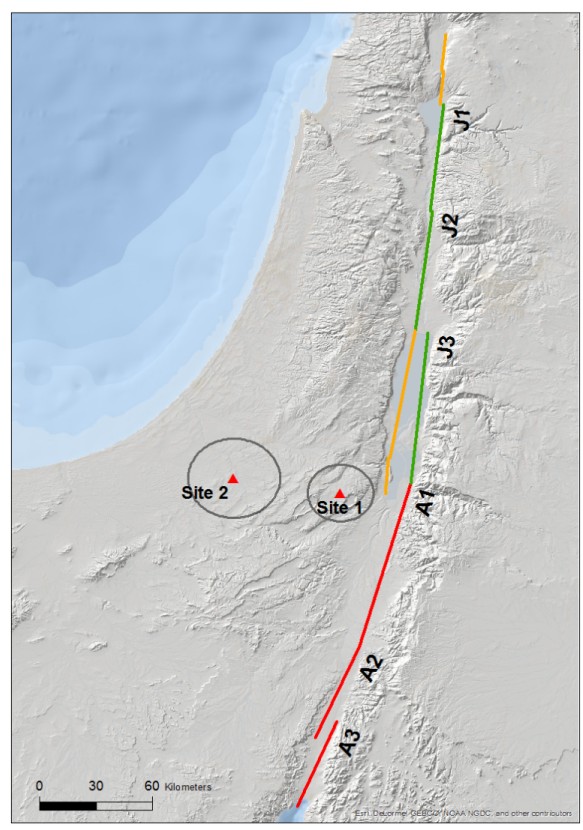

**Figure 6. The segmented model of the DST, in which the Arava and Jericho are each represented by three segments. Arava is represented by the red faults numbered A1 through A3. Jericho is represented by the green faults numbered J1 through J3. Orange segments were not included in the analysis herein, to avoid double-counting of moment release.**





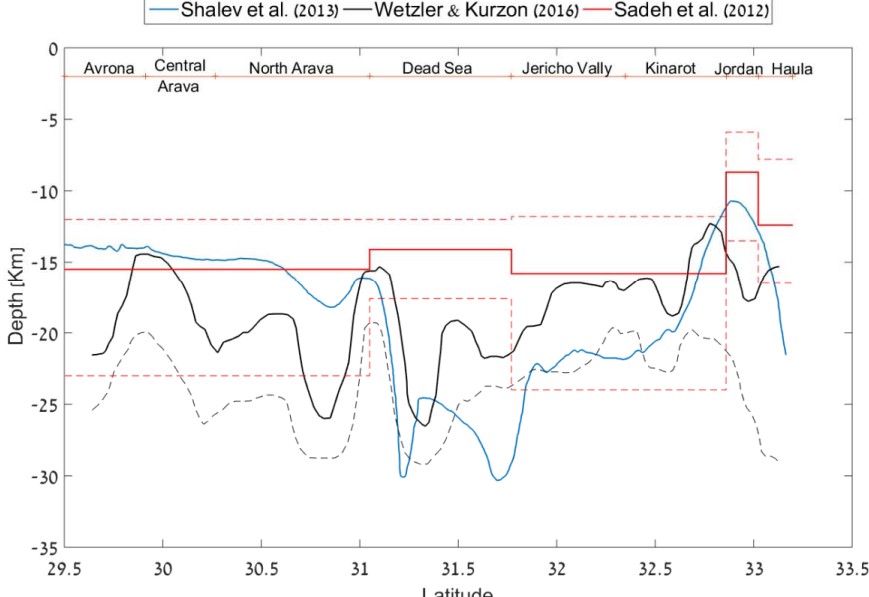

**Figure 7. The range of evaluations for the seismogenic crustal depth along the DST, based on three independent studies. The X-axis represents a cross section along the DST – from South (left) to North (right). The figure is drawn at a vertical exaggeration of 11. In the Wetzler and Kurzon (2016) study – the solid line represents the 75[th] percentile while the dashed line represent the 95[th] percentile. In the Sadeh et al. (2012) study – the solid line represents the estimated depth with a confidence level of 68%, while the dashed lines represent two standard-deviations above and below that estimate.**





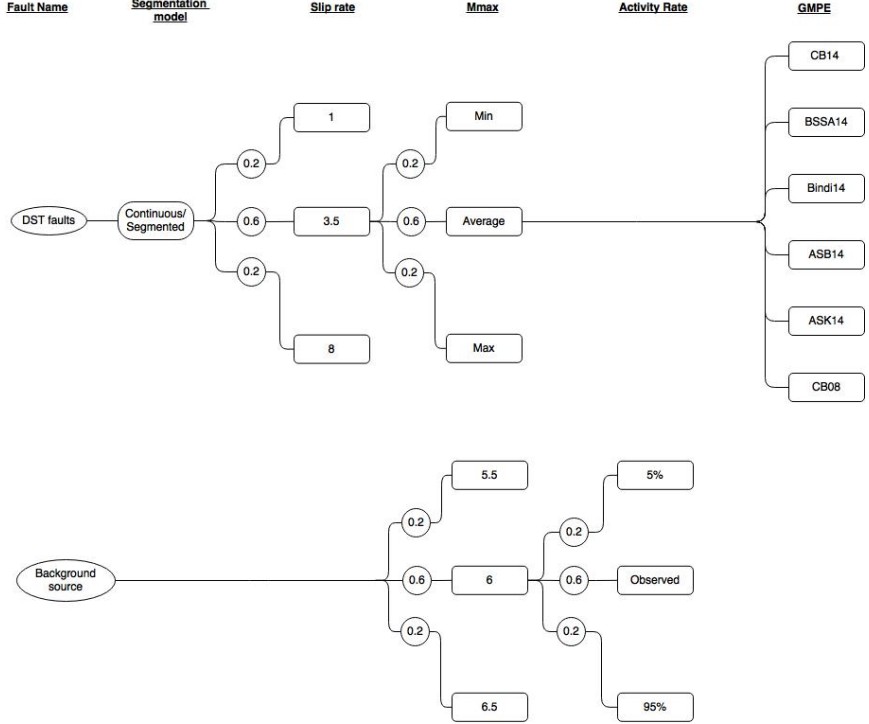

**Figure 8. Logic tree for Model 5, showing the main branches of the linear sources (top) and the background polygon (bottom). Where a weight isn't assigned, the analysis was conducted separately for each alternative.**





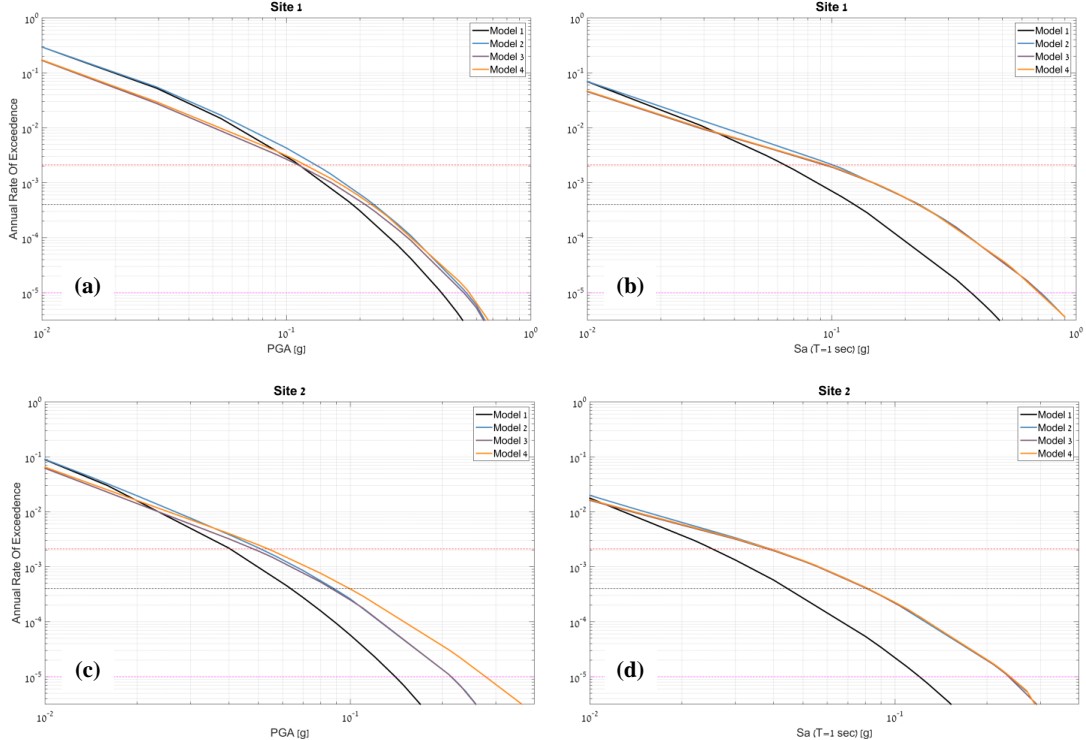

**Figure 9. Hazard curves obtained using Models 1 through 4 for (a) Site #1 at T=0.01sec, (b) Site #1 at T=1.0sec, (c) Site #2 at T=0.01sec, and (d) Site #2 at T=1.0sec.**





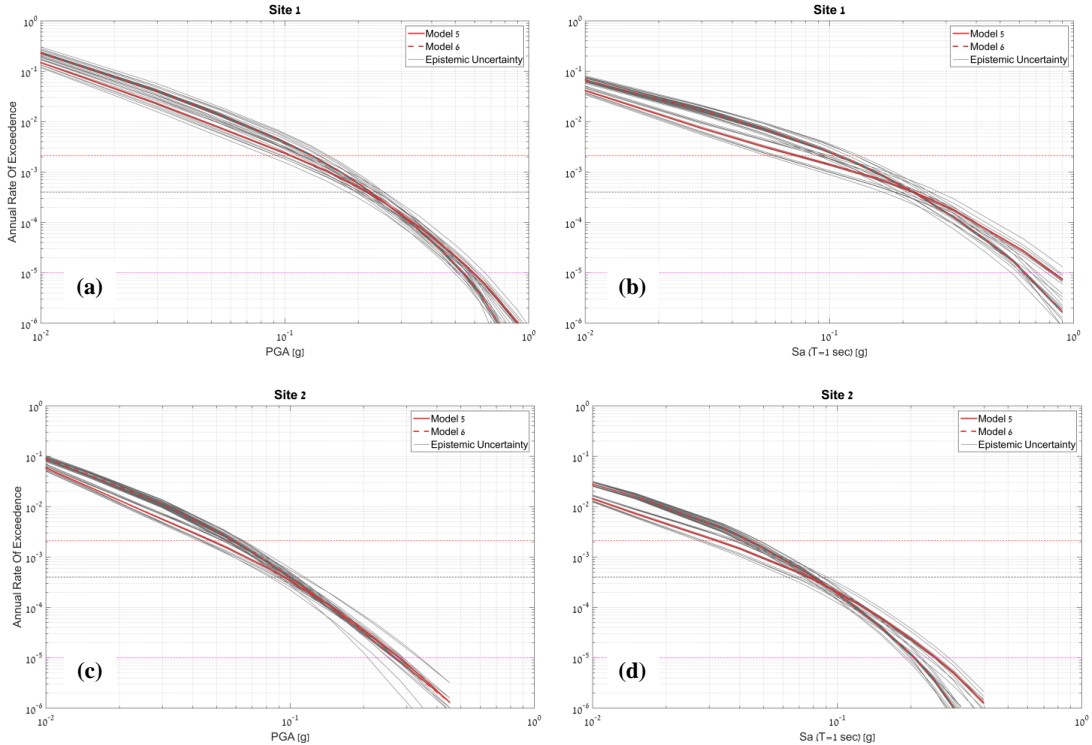

**Figure 10. Hazard curves for Models 5 and 6, showing the full range of parametric uncertainty, for (a) Site #1 at T=0.01sec, (b) Site #1 at T=1.0sec, (c) Site #2 at T=0.01sec, and (d) Site #2 at T=1.0sec.**





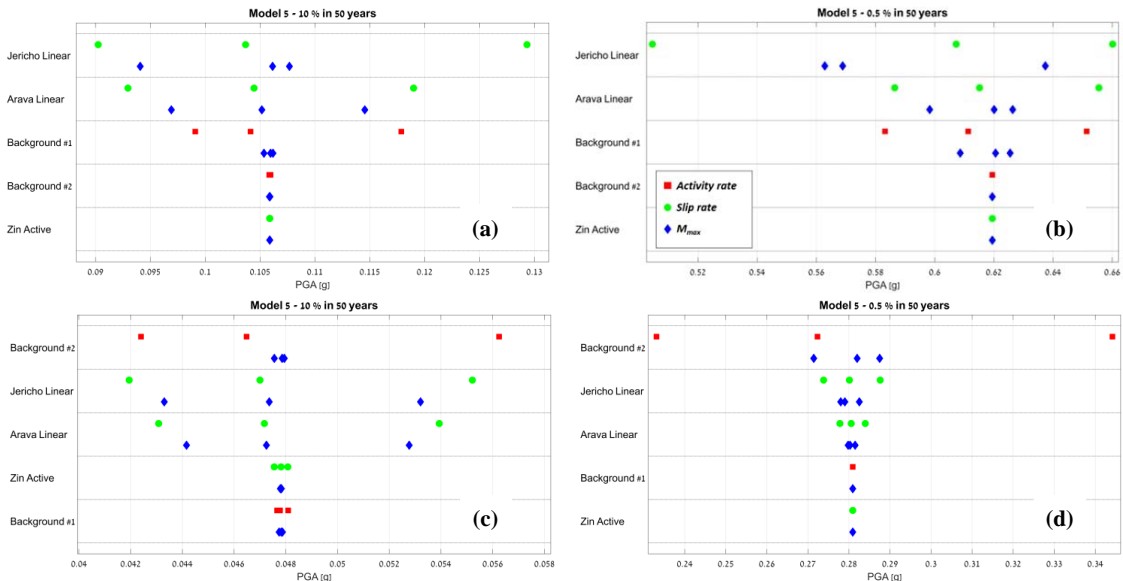

5    **Figure 11. Tornado plots for PGA only, showing the contribution of parametric uncertainty to the hazard for (a) Site #1, 10%@50 yrs, (b) Site #1, 10E⁻⁵, (c) Site #2, 10%@50 yrs, and (d) Site #2, 10E⁻⁵ .**

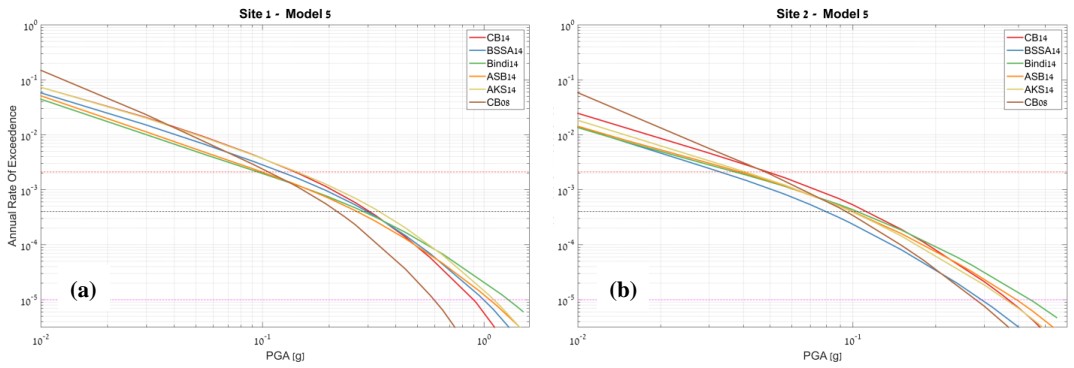

**Figure 12. Hazard curves for Model 5, using the average weighted values, at (a) Site #1, and (b) Site #2.**




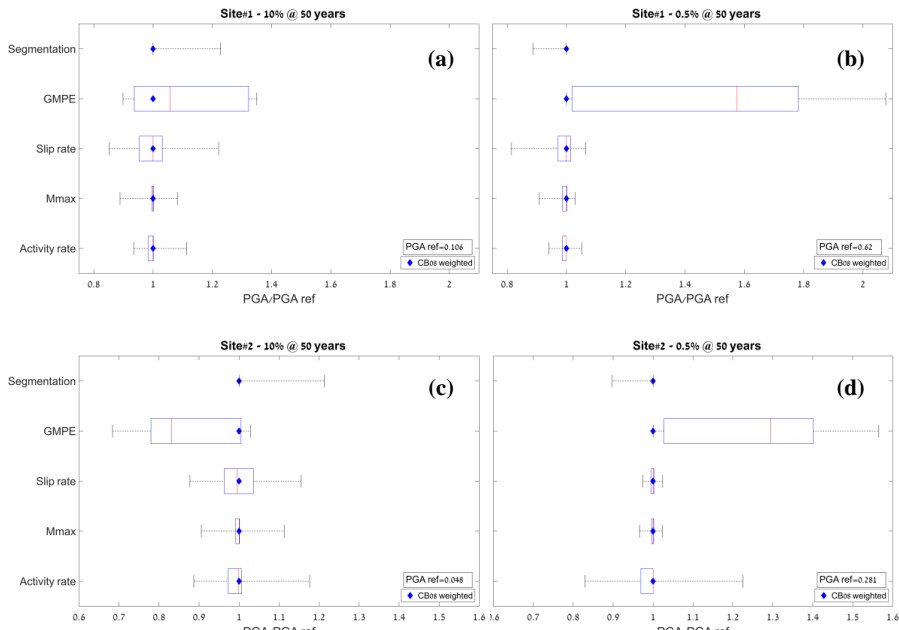

**Figure 13. Summary plots, showing the overall relative effect of the uncertainty associated with four different parameters on hazard
5  results using Model 5. Site #1 (top) vs. Site #2 (bottom), at two different recurrence intervals – 1/475 yr (left) vs. 1/10$^5$ yr (right). The
blue diamond is the weighted average of Model 5 using the CB08 GMPE. The red line is the median value and the box represents the
25$^{th}$ and 75$^{th}$ percentiles of the results.**