# Peer review of "The Effect of Alternative Seismotectonic Models on PSHA Results – a Sensitivity Study for Two Sites in Israel"

_Natural Hazards and Earth System Sciences, 2017_

## Referee Comment (RC1) · Anonymous Referee #1 · 7 Sep 2017

This is a really well-written manuscript. Don't have any suggestions to further improve it.

---

## Referee Comment (RC2) · Anonymous Referee #2 · 2 Oct 2017

Comments on the manuscript "The Effect of Alternative Seismotectonic Models on PSHA Results – A Sensitivity Study for the Case of Israel" by Matan, Michael, Ory and Ronnie to be published on NHESS journal

This and similar studies are important and necessary for assessment of potential seismic hazard in Israel as well as the areas with high and low seismic activity. This study presents the alternative seismotectonic and earthquake rate models and their use in PSHA. It performs the seismogenic zonation approach as well as the characteristic earthquake approach as an alternative using the updated seismic and geological database. Moreover, it deals with some parametric sensitivity analysis which may be

used to improve the probabilistic seismic hazard assessment (PSHA) for future seismic hazard assessment in Isreal. The uncertainty analysis suggests that future research is necessary to resolve some of the key issues that affect earthquake occurrence probabilities and ground shaking hazard. Therefore, the topic could be of interest for the readers of NHESS, however, due to several important points discussed below, the manuscript requires some revisions and improvements. The use of English language is well however very colloquial at some parts. The main problem concerns completely missing definition of the whole uncertainty procedure followed in the study in particular, lack of the information and the methodology adopted/performed for the sensitivity analysis as well as the raking procedure related to the seismic sources. I recommend authors including a new section where they explain the uncertainty as well as the sensitivity analysis together with proper equations and clear definitions. 1- In section 2 authors define the range of parametric epistemic uncertainties cautiously (pag 4,5,6,7 and 8) then they suddenly start presenting the results (pag.8, 9 and 10) without given any methodology/approach that follow to obtain those uncertainties. For example figure 11 and figure 13 needs to be explained in a better way. 2- In figure 9 authors presenting hazard curves that are calculated at two sites using four models for two spectral periods (Table1), but there is no uncertainty study dedicated to this section; figure simply presents four hazard curves. Some quantitate presentation and values are needed in order to highlight the differences between the areal sources and the linear fault one (S1413 and DD14); for example what is the percentile change or the absolute peak ground acceleration values. Please also indicate adopted GMPEs for the hazard analysis for these four models. 3-It is not clear to me how the authors utilized the logic tree for the sensitivity study, please indicate clearly how the selected weights (figure 8) associated to each branch are treated during the analysis. How the seismic sources are ranked by their contribution to the hazard uncertainty such that those of most contributing. How many model and parameter combinations did you realized? Here needs some quantitative values. 4-Please explain the use of the three standard deviations in the hazard calculations (Page9 pr.35-40)? 5-In Figure 13 explain

why 25th and 75th percentiles are different around the median values? 6-There are too many input data, parameters and/ or database are mostly acknowledged without given any explanation. For example, how the seismic catalog were pre-processed (completeness analysis, declustering, Magnitude types etc.) and how many event does the catalog contain as a function of magnitude should be given for the benefit of readers. It is fundamental that the single seismicity parameter calculated (b-value, recurrence times for each fault) and assumed during the model construction expressed clearly with the complete and proper equation (use of YC, TN, TE) definition and description. 7-Section 2.1.4 activity rates calculated for 2 sites and given in table 3, but how the Weichert (1980) approach performed using GJI catalog is not clear! as my knowledge the method takes into account the completeness magnitude thresholds by various time periods, I was wondering how authors benefit this to obtain long-term activity rate in the region. Since it is one of the most important parameters it merits to be explained in a better way. Again please present what is the b-value/s calculated for the study area/s. 8-Authors state that the use of slip rates as taken 1-8 mm/yr range in their study but it is not clear how and where they use this information is totally missing (page 5-6). Similarly in the segmentation model, authors claimed that only the eastern segment was chosen to represent the faulting in the Dead Sea basin in order to maintain the correct moment balance but there is no equation, figure, quantitative value that demonstrates this choose (page 6 par.15). Same as the magnitude calculations from different fault segments needs more information and more scientific definitions (page 7 par.10-15). How the fault's depths are taken into account for the Mmax calculation; more information and/or suitable set of equations needs to be provided. There is no information regarding to fault type and the mechanisms (strike-slips or normal faulting?). 9-What are the weights used for 6 GMPEs through the logic tree. 10-Most of the maps are not informative and not always show explicitly city and county/sea names. Figure captions must be more informative. 11-Please insert the seismicity of region into figure 1-2 and 4 12-Abstract: it is necessary describing the important quantitative results obtained in the study. 13-In the discussion section please avoid the colloquial presentations as

under- or overestimates instead present these variabilities with quantitative values.

---

## Author Comment (AC1) · 13 Nov 2017

Thank you for the compliment, no further revisions required.
* * *

---

## Author Comment (AC2) · 13 Nov 2017

**The Effect of Alternative Seismotectonic Models on PSHA Results – a Sensitivity Study for the Case of Israel**

By: Avital, Kamai, Davis, and Dor.

**Response to Reviewer #2**

We are thankful to the reviewer for his/her valuable and constructive comments. In the revision, we have taken into account all the comments and made changes accordingly. Details of the actions taken regarding the comments and edits are provided below. All comments are in italics, with the corresponding replies listed directly below.

**Response to RC2:**

*The main problem concerns completely missing definition of the whole uncertainty procedure followed in the study in particular, lack of the information and the methodology adopted/performed for the sensitivity analysis as well as the raking procedure related to the seismic sources. I recommend authors including a new section where they explain the uncertainty as well as the sensitivity analysis together with proper equations and clear definitions.*

As per the reviewers' suggestions – we added more details about the analysis, as specifically explained in response to the other comments below. However – a full section with equations and definitions was not added for the following reason: this paper is intended for an audience who is familiar with the methodology of PSHA. It is beyond the scope of the paper to explain definitions that are trivial to hazard analysts throughout the world, because (a) those equations and definitions are already fully presented and explained elsewhere, and (b) it will make the current manuscript long and tedious.

More importantly: The purpose of this study is to answer the following question:   "how does the epistemic uncertainty, associated with source and path parameters, affect hazard results in Israel". The paper takes two sites as an example, but it is clear that the exact values (by numbers) will vary between sites, while the main trends will likely be consistent. Therefore, we see no point in exhausting the reader with numbers that are only true to the two sites we tested. We focus on trends and the scale of the effect so that the main conclusion isn't lost within the numbers.

*In section 2 authors define the range of parametric epistemic uncertainties cautiously (pag 4,5,6,7 and 8) then they suddenly start presenting the results (pag.8, 9 and 10) without given any methodology/approach that follow to obtain those uncertainties. For example figure 11 and figure 13 needs to be explained in a better way.*

We accept the reviewer's comment - detailed explanations regarding Figures 10 amd 11 were added to the text. Other than that, there is not much detail to add regarding the uncertainty analysis, because the statistical analysis itself is very basic and it is mostly a question of how to visualize the range of results obtained by running multiple hazard scenarios.

*In figure 9 authors presenting hazard curves that are calculated at two sites using four models for two spectral periods (Table1), but there is no uncertainty study dedicated to this section; figure simply presents four hazard curves. Some quantitate presentation and values are needed in order to highlight the differences between the areal sources and the linear fault one (S1413 and DD14); for example what is the percentile change or the*

*absolute peak ground acceleration values. Please also indicate adopted GMPEs for the hazard analysis for these four models.*

As suggested by the reviewer – numerical information was added to the paragraph discussing Figure 9, in order to highlight the differences. For example – for site #2, PGA, long return intervals – the difference between Model 1 and 2 ranges between 25% and 52% for short and long return intervals, respectively. The difference between model 3 and 4 ranges from 10% to 30% for the short and long return intervals, respectively.

A Clarification has been added to the first paragraph of the results chapter:" Note that all hazard analyses are performed with the CB08 GMPE, unless specified otherwise".

*It is not clear to me how the authors utilized the logic tree for the sensitivity study, please indicate clearly how the selected weights (figure 8) associated to each branch are treated during the analysis. How the seismic sources are ranked by their contribution to the hazard uncertainty such that those of most contributing. How many model and parameter combinations did you realized? Here needs some quantitative values.*

A clarification has been added to the paragraph explaining Figure 10: "The weighted average, calculated using the logic tree weights, as shown in Figure 8, is represented in Figure 10 by the solid and dashed red lines, for models 5 and 6, respectively".

Figure 10 includes 126 realizations of hazard curves. This information was also added to the respective paragraph in the main text.

*Please explain the use of the three standard deviations in the hazard calculations (Page9 pr.35-40)?*

Truncating the PSHA analysis at three standard deviations is a typical practice in PSHA. Bommer and Abrahamson (2006) show that truncating at a lower level violates the inherent data variability, while truncating at a higher level has little effect on the results. In the main text, we added to line 41 in page 9: "..as typically done in PSHA practice (Bommer and Abrahamson, 2006)".

*In Figure 13 explain why 25th and 75th percentiles are different around the median values?*

The $25^{th}$ and $75^{th}$ percentiles should only be symmetric around the median in a normally-distributed sample. Here, we do not necessarily deal with normally-distributed data, hence the percentiles are not symmetric. This is why we show the percentiles and not the standard deviation, which is only correct for normally-distributed samples.

*There are too many input data, parameters and/ or database are mostly acknowledged without given any explanation. For example, how the seismic catalog were pre-processed (completeness analysis, declustering, Magnitude types etc.) and how many event does the catalog contain as a function of magnitude should be given for the benefit of readers. It is fundamental that the single seismicity parameter calculated (b-value, recurrence times for each fault) and assumed during the model construction expressed clearly with the complete and proper equation (use of YC, TN, TE) definition and description.*

The seismic catalog was not processed at all for this work. All seismicity values, apart for the two background zones presented in Figure 4, were used from the literature. That is why we do not report them, or explain how they were calculated. That information is already reported elsewhere, and is not part of this study. The only seismic parameters we calculate in this study are the activity rates of the background zones, and those are based on less than five earthquakes each, so no declustering is necessary. Completeness of the GII catalog has been discussed in previous reports.

*Section 2.1.4 activity rates calculated for 2 sites and given in table 3, but how the Weichert (1980) approach performed using GJI catalog is not clear! as my knowledge the method takes into account the completeness magnitude thresholds by various time periods, I was wondering how authors benefit this to obtain long-term activity rate in the region. Since it is one of the most important parameters it merits to be explained in a better way. Again please present what is the b-value/s calculated for the study area/s.*

Table 1 in Weichert (1980) suggests lower and upper factors for calculating confidence intervals on the Poisson's Mean N, for small N, when N is the number of events. Meaning – if the measured N is a small number, we cannot count on it being normally-distributed, and hence need to use a Chi-squared distribution and equation (11) in the Weichert paper. This is commonly done when evaluating activity rates for non-seismically active areas. The b-value is the same for the entire country, as explained in the literature.

*Authors state that the use of slip rates as taken 1-8 mm/yr range in their study but it is not clear how and where they use this information is totally missing (page 5-6). Similarly in the segmentation model, authors claimed that only the eastern segment was chosen to represent the faulting in the Dead Sea basin in order to maintain the correct moment balance but there is no equation, figure, quantitative value that demonstrates this choose (page 6 par.15). Same as the magnitude calculations from different fault segments needs more information and more scientific definitions (page 7 par.10-15). How the fault's depths are taken into account for the Mmax calculation; more information and/or suitable set of equations needs to be provided. There is no information regarding to fault type and the mechanisms (strike-slips or normal faulting?).*

Slip rate – range used for DST segments and associated weights are given in the logic tree. Figure 8.

Segmentation – we changed the text as follows: "In order to maintain the correct moment balance in the segmented model (i.e. maintain the total fault length), only the eastern segment was chosen to represent the faulting in the Dead-Sea basin. This is consistent with findings from Sadeh et al. (2012), who show that most of the movement occurs on the eastern segment of the Dead-Sea basin fault."

Mmax – we added a clarification that all DST faults are assumed to be strike-slip with a 90 degree dip. This is a transform fault, so this assumption is very likely correct.

Mmax is calculated using Hanks and Bakun model (2002), which is very well-known in the field. That is why we choose to avoid adding unnecessary equations.

*What are the weights used for 6 GMPEs through the logic tree.*

The logic tree is only used to compute the average hazard curve for Models 5 and 6, as presented in Figure 10 by the red lines. The GMPEs are each computed separately, using the average Model 5, but 100% for each GMPE in each run (i.e. no weights). In page 9, line 36 we specifically relate to the 'weighted average of Model 5' in Figure 12. A clarification has been added in the text right after Table 4.

*Most of the maps are not informative and not always show explicitly city and county/sea names. Figure captions must be more informative. Please insert the seismicity of region into figure 1-2 and 4*

Following the reviewers comment – all boundary coordinates and seismicity parameters for both models #1 and #3 are now available as supplementary material in electronic appendices E1 and E2, respectively. A note has been added to the figure captions of Figure 1 and 2. We avoid political boundaries in this region because they are debatable.

*Abstract: it is necessary describing the important quantitative results obtained in the study.*

*In the discussion section please avoid the colloquial presentations as under- or overestimates instead present these variabilities with quantitative values.*

Quantitative values have been added to the abstract.

**References**

Bommer, J.J., Abrahamson, N.A., 2006. Why do modern probabilistic seismic-hazard analyses often lead to increased hazard estimates? Bulletin of the Seismological Society of America 96, 1967-1977.

---

## Editor Decision (ED1)

[revised manuscript text omitted]

REFERENCED PAPERS SHOULD BE ADDED
OR QUOTE THE WORK WHERE THIS PICTURE
IS TAKEN AND/OR MODIFIED

[Figure]

*(handwritten annotations:)* → NO CLEAR THE SEGMENTATION A1–A2

*faults*

*lines*

Figure 6. The segmented model of the DST, in which the Arava and Jericho are each represented by three segments. Arava is represented by the red faults numbered A1 through A3. Jericho is represented by the green faults numbered J1 through J3. Orange segments were not included in the analysis herein, to avoid double-counting of moment release.

[Figure]

**Figure 7. The range of evaluations for the seismogenic crustal depth along the DST, based on three independent studies. The X-axis represents a cross section along the DST – from South (left) to North (right). The figure is drawn at a vertical exaggeration of 11. In the Wetzler and Kurzon (2016) study – the solid line represents the 75[th] percentile while the dashed line represent the 95[th] percentile. In the Sadeh et al. (2012) study – the solid line represents the estimated depth with a confidence level of 68%, while the dashed lines represent two standard-deviations above and below that estimate.**

WHAT ABOUT SHALEV ETAC 2013?

[Figure]

**Figure 8. Logic tree for Model 5, showing the main branches of the linear sources (top) and the background polygon (bottom). Where a weight isn't assigned, the analysis was conducted separately for each alternative.**

The representation of continuous / segmented model has to be split in seperate boxes, the tree of one branch only scratched by dots

[Figure]

*BIGGER FONT* (handwritten annotation)

**Figure 9. Hazard curves obtained using Models 1 through 4 for (a) Site #1 at T=0.01sec, (b) Site #1 at T=1.0sec, (c) Site #2 at T=0.01sec, and (d) Site #2 at T=1.0sec.**

*Short spectral period T=0.01s is referred as PGA in the text* (handwritten annotation)

*"s" not "sec"* (handwritten annotation)

BIGGER FONT

[Figure]

Figure 10. Hazard curves for Models 5 and 6, showing the full range of parametric uncertainty, for (a) Site #1 at T=0.01sec, (b) Site #1 at T=1.0sec, (c) Site #2 at T=0.01sec, and (d) Site #2 at T=1.0sec. The weighted averages are represented by the red curves.

use "s" not "sec"

[Figure]

Figure 11. Tornado plots for PGA only, showing the contribution of parametric uncertainty to the hazard for (a) Site #1, 10%@50 yrs, (b) Site #1, $10E^{-5}$, (c) Site #2, 10%@50 yrs, and (d) Site #2, $10E^{-5}$ .

*(handwritten annotations):* 0.5% @ 50 yrs

*for Model 5 BIGGER FONT*

[Figure]

Figure 12. Hazard curves for Model 5, using the average weighted values, at (a) Site #1, and (b) Site #2.

[Figure]

*(Handwritten annotations: "BIGGER FONT" with circle around "PGA ref=0.62  CB08 weighted" legend in panel (b))*

**Figure 13.** Summary plots, showing the overall relative effect of the uncertainty associated with four different parameters on hazard results using Model 5. Site #1 (top) vs. Site #2 (bottom), at two different . The blue diamond is the weighted average of Model 5 using the CB08 GMPE. The red line is the median value and the box represents the 25$^{th}$ and 75$^{th}$ percentiles of the results.

*(Handwritten annotation: "return period (475 yrs left, 100,000 yrs right)")*

---

## Author Response (AR2)

**The Effect of Alternative Seismotectonic Models on PSHA Results – a Sensitivity Study for the Case of Israel**

**By: Avital, Kamai, Davis, and Dor.**

**Response to Editor**

I thank the editor for her close review of the paper and detailed comments. I did my best to address the comments, although I must admit it was very difficult to understand the hand writing in some places so I may have misunderstood the intention. Ronnie Kamai.

*I propose you a modification of the title and minor changes in the abstract*

Most changes have been adopted, apart for one sentence I could not understand due to handwriting.

*I ask you to modify the organization of the contents of introduction, moving the blocks of text now titled 1.2 and 1.3 after the first paragraph, page 1 line 28. Avoid subtitle if not absolutely necessary, move block 1.1 in the opening of chapter 2*

Accepted and changed.

*Reformat the descriptions of models in a bulleted list, if you want to separate parametric from modelling uncertanty use the second level title 2.1, otherwise upgrade 2.1.1 to 2.1 and so on*

Accepted and changed. Subtitle for modelling and parametric epistemic uncertainty maintained, to provide clear definition to readers who are less familiar with the distinction

*Pages 9 to 11 have sometimes contradictions, it is difficult to understand why the logic tree is one branch only for some models, and the conclusions you derive from that. Perhaps explaining the reasons why you discarded the esploration of all the branches will help the reader to understand*

All branches were explored. We simply did not extend them on the logic tree schematics, to avoid over-cluttering. Dotted lines have been added now to illustrate this point. The segmentation branches do not have weights because we did not average between the two scenarios.

*Figures have to be more readable in legends, avoid titles out of the frame of each panel, check the use of abbreviations*

Comments in the annotated file have been addressed.

*All the references marked with a double circles do not follow the standards of the journal, please check them and provide evidence of the restricted materials*

References have been corrected, where possible. Some of the reference are older reports in Hebrew, for which report number, or publisher, are not always available..

*I suggest that you quote explicitely the b-value used in Table 1, distance metrics adopted in Table 4.*

We do not explicitly quote the b-values in Table 1, because there is more than 1 value per model. There are 3 different b-values in Model 1, which are maintained across the other models and specified in the electronic appendix. We did, however, add the distance metrics to Table 4, as suggested.

**The Effect of Alternative Seismotectonic Models on PSHA Results – a Sensitivity Study for Two Sites in Israel**

Avital, Matan[1], Kamai, Ronnie[2], Davis, Michael[3] and Dor, Ory[3]

[1]Department of Geological and Environmental Sciences, Ben-Gurion University of the Negev, Beer-Sheva, 84105, Israel
[2]Department of Structural Engineering, Ben-Gurion University of the Negev, Beer-Sheva, 84105, Israel
[3]Ecolog Engineering, Inc. Rehovot, 7670203, Israel

*Correspondence to*: Ronnie Kamai (rkamai@bgu.ac.il)

**Abstract**

We present a full Probabilistic Seismic Hazard Analysis (PSHA) sensitivity analysis for two sites in southern Israel – one in the near-field of a major fault system and one farther away. The PSHA analysis is conducted for alternative source representations, using alternative model parameters for the main seismic sources, such as slip-rate and $M_{max}$, among others. The analysis also considers the effect of the Ground-Motion Prediction Equation (GMPE) on the hazard results. In this way, the two types of epistemic uncertainty – modelling uncertainty and parametric uncertainty – are treated and addressed. We quantify the uncertainty propagation by testing its influence on the final calculated hazard, such that the controlling knowledge gaps are identified and can be treated in future studies. We find that current practice in Israel, as represented by the  current version of the building code, grossly underestimates the hazard, by approximately 40% at short return periods (e.g. 10% in 50 yrs) and by as much as 150% at long return periods (e.g. 10E-5). The analysis shows that this underestimation is most probably due to a combination of factors, including source definitions as well as the GMPE used for analysis.

**1    Introduction**

Israel lies on an active plate boundary, —with the Dead-Sea Transform (DST) separating the African plate on the west from the Arabian plate on the east. According to the historical, biblical, and archaeological records (Ben-Menahem 1991), devastating earthquakes with recurrence intervals of approximately 100 years are responsible for the repeated destruction of cultural centres in this region. While Israel benefits from a relative wealth of historical, geological and paleoseismological dataset that can supports Seismic Hazard Assessments (SHA), its instrumental catalogue is poor due to the combination of its young age, sparse spatial coverage, and moderate seismicity rates. Therefore, the current state-of-practice for conducting seismic hazard analysis in Israel suffers from some significant knowledge gaps and methodological shortcomings, which may lead to erroneous hazard estimations.

The most recent update to the Israeli building code (SII 2013) and its associated seismic-hazard map (Klar et al. 2011) is considered herein to represent the state-of-practice of seismic hazard analysis in Israel. This practice will be further related to herein as the 'SI413' model. The underlying seismotectonic model in SI413 is shown in Figure 1. It is composed of areal sources only, based on the work of Shamir et al. (2001). The activity rates within the seismic zones were defined based on the uniform earthquake catalogue, constructed from combined historical and instrumental data (Shapira and Hofstetter 2002, Shapira et al. 2007). The seismic zones are all assigned a truncated exponential (TE) magnitude-frequency distribution (MFD), as typical for

**Comment [RK1]:** Editor comment: change to PSHA.
Response: This data can be used for both deterministic and probabilistic, and therefore the use of SHA as a general term is maintained.

[revised manuscript text omitted]